# Decreased sound tolerance in a Canadian University Context: Associations with autistic traits, social competence, and gender in an undergraduate sample

**Silas Manning, Natalia Van Esch, Nichole E. Scheerer** [iD] *

Department of Psychology, Wilfrid Laurier University, Waterloo, Ontario, Canada

* nscheerer@wlu.ca

## Abstract

Disorders of decreased sound tolerance such as misophonia and hyperacusis cause significant distress through strong negative emotional and physiological reactions to everyday sounds. These conditions have been associated with poor mental and physical health as they impact day to day life. Prior to the recent development of consensus definitions of misophonia and hyperacusis, attempts to determine the prevalence of these conditions have been hindered by the ambiguity and inconsistency of their descriptions. Despite this, certain populations have been suggested to more frequently experience misophonia and hyperacusis, namely younger people and autistic people. Furthermore, there has been conflicting evidence regarding whether these conditions are more prevalent among women. Post-secondary campuses are often sensory-rich. As such, students with misophonia and hyperacusis are likely to experience distress. For this reason, we sought to investigate the prevalence of misophonia and hyperacusis in a Canadian university sample, and explore the relations between these conditions and gender, autistic traits, and overall social competence. As exposure to many everyday sounds can be highly aversive for individuals with misophonia and/or hyperacusis, these individuals often attempt to avoid environments in which distressing sounds are encountered. It is therefore possible that poorer social competence may be a secondary effect of these conditions. To investigate these relations, 2080 students completed an online survey using multiple established self-report measures of decreased sound tolerance. Clinical misophonia was detected in 12–18% of participants, and hyperacusis in 6–17%. Both conditions were significantly more prevalent among women than men. Both conditions were found to be weakly to moderately positively correlated with autistic traits, and weakly to moderately negatively correlated with social competence. These results highlight decreased sound tolerance as a significant issue at Canadian post-secondary institutions, calling for steps to be taken to mitigate its effects.

**Data availability statement:** All relevant data are within the paper and its Supporting Information files.

**Funding:** This research was supported by the Wilfrid Laurier University Research Support fund that helps to support all faculty and students at Laurier (i.e., SM, NV, NS). There is no specific grant number. The funders had no role in study design, data collection and analysis, decision to publish, or preparation of the manuscript. There was no additional external funding received for this study.

**Competing interests:** The authors have declared that no competing interests exist.

## Introduction

Decreased Sound Tolerance (DST) is an umbrella term for conditions in which a person has an elevated sensitivity to sound in forms that would not bother most people, such as misophonia and hyperacusis [1–3]. People with DST experience a host of aversive reactions to sound including fear, pain, and anger [2,3]. DST can also lead to problematic behaviors including avoidance [2] and, in the case of misophonia, self-harm [4]. Given the potential of DST to interfere with academic and social functioning, as well as overall well-being, the aim of this current research is to elucidate the prevalence of two classes of DST, misophonia and hyperacusis, in a large sample of university students. Because DST often co-occurs with autism [5,3], a condition related to differences in social behaviour [6], we will also explore the relationship between autistic traits, social competence, and DST prevalence in this undergraduate sample. Better characterizing DST in post-secondary students will set the stage for increased recognition and accommodation of DST in post-secondary environments, and more broadly.

### Misophonia

First described by Jastreboff and Jastreboff in 2001, misophonia is a form of DST characterized by a strong negative reaction, such as panic or rage, elicited by sound "triggers" and the stimuli associated with them [1,7]. Until recently, the lack of consensus on a specific clinical definition of misophonia has slowed research on this disorder, as inconsistent criteria make comparisons across different studies especially complicated. However, in 2022 a Delphi study was conducted to develop a definition of misophonia by consensus [7]. In brief, the consensus definition states that misophonia is a disorder of decreased tolerance to specific sounds or stimuli associated with those sounds. The stimuli in this case, known as "triggers", are perceived as highly unpleasant and evoke a strong, negative emotional and/or behavioural response. Triggers are often repetitive, and frequently consist of sounds made by other individuals, especially human produced sounds such as chewing, sniffing, or other mouth sounds. Once detected, it can be very difficult for a person to distract themselves from the triggering stimulus and this can increase distress. Over time this can lead to impairment in social, occupational, or academic functioning [7].

This definition, along with supporting research on the age of onset, symptom severity, and family history, provides evidence that misophonia is a distinct condition, rather than a symptom of other psychiatric, audiological, or medical conditions as once suggested [8,9]. While this definition represents a significant step towards a unified understanding of misophonia, it is not without its critics and will continue to be updated and improved as more research is conducted.

In misophonia, exposure to trigger sounds can evoke physiological symptoms such as sweating and increased heart rate [10]. Strong behavioural reactions can be present as well, such as aggression directed at the individual generating the triggering sounds [7]. Avoiding or escaping situations in which triggers may be encountered, and making attempts to halt the stimuli, are common measures taken to cope with

the response [2,7,11]. The social context in which the trigger is encountered has also been shown to play a role in the severity of the reaction to the trigger [7,12].

## Hyperacusis

Hyperacusis, another form of DST, is characterized by a similar strong negative reaction to sound. Hyperacusis has also recently been examined by a group of researchers as part of a Delphi consensus study [13]. This resulted in a definition in which hyperacusis is a reduced tolerance to normal, everyday sounds as they are perceived as substantially louder than would be perceived by a person without hyperacusis. This heightened perceived volume of sounds can be unpleasant, overwhelming, and/or painful [14]. Thus, the impact of hyperacusis is similar to that of misophonia, though the source of the reaction is less specific. Hyperacusis is frequently comorbid with tinnitus, with many reporting hyperacusis as a secondary complaint to tinnitus [15]. In contrast to tinnitus wherein issues usually relate to hearing a restricted frequency range, hyperacusis is thought to relate to a generalized increase in auditory gain, as loudness discomfort levels are lower in hyperacusis across a broad range of auditory frequencies [15].

## Impact of DST

The experience of a person with DST can present in many ways, for example pain in response to the sound of a laptop fan, or rage or panic upon exposure to the sounds of a close friend or relative eating. Given DST triggers are so common in everyday environments, living with DST can have a substantial influence on the overall wellbeing of those affected [2,5,13]. Specifically, DST has been found to be related to high stress as well as anxiety and depression [4,5,12,16]. This pattern is similar to what is observed in many chronic health issues, as the constant barrage of mildly or strongly aversive events impacts an individual over time. As university involves a high-stress, sensory-rich environment where trigger sounds can be difficult to avoid, DST conditions such as misophonia and hyperacusis may create significant barriers to academic success for students with these conditions.

Since individuals with misophonia and hyperacusis commonly engage in methods to avoid or reduce exposure to their triggers, this can mean withdrawing from certain places and/or groups of people [2]. As such, DST has been reported to impact relationships with family and friends, harm performance at school or work, and can generate a fast track to other conditions that worsen quality of life even further [2,8,17,18]. At least one study on misophonia's impact found that for teenagers and adults with misophonia, there is a significantly elevated rate of both self-harm behaviours and suicidal ideation, particularly in women and girls [4]. It is therefore clear that these are serious conditions that necessitate attention and care to avoid dire consequences.

In Canada, a typical post-secondary education such as an undergraduate degree at a university involves a substantial amount of time spent in a highly varied set of sensory-rich environments. For example, lecture halls often contain laptops, cellphones, projectors, air conditioning systems, food, drinks and of course people – all producing sources of sensory stimulation. On-campus residences will naturally involve groups of people cohabiting in close proximity. Dining halls, food courts, outdoor common areas, athletic centres, and more, all offer unique sensory experiences. For individuals living with DST, these everyday sensory-rich environments mean they are highly likely to encounter triggering stimuli. As a result, the impact of one's DST condition(s) is likely to be felt every single day unless there is some form of intervention. The simplest option for mitigating this exposure to triggering stimuli is to avoid and/or withdraw from triggering environments. However, this strategy is highly antithetical to succeeding academically and socially during this important developmental period. It is therefore clear that solutions need to be available for students with misophonia and/or hyperacusis to navigate their time at post-secondary institutions without the frequent negative experiences brought about by their negative reactions to stimuli.

In general, past research has shown that elevated psychological distress has long been a major factor in the lives of university student populations worldwide as there are many things for students to be stressed about during this period

[19], so naturally the presence of one or more DST conditions adds yet another layer of distress. Crucially, as misophonia most often emerges in adolescence [20], those with severe enough cases to warrant frequent social withdrawal could be missing out on key elements of social development, potentially leading to reduced social competence and further worsening mental health. It is widely known that mental health issues are a major concern for both students and university administrations, and while a review of mental health services at Canadian universities found that reasonably good services are present at many institutions, there remain barriers to access as well as inadequate resources in many cases, resulting in a difficult path for many students that seek help [21]. Unfortunately, despite similar studies on depression, attention deficit hyperactivity disorder (ADHD), and autism among others, research on how DST specifically impacts physical and mental health, and student success, while studying at a post-secondary institution, has not yet been conducted.

## Prevalence estimates

An estimation of the prevalence of DST is a crucial early step to focus attention where it is most needed. Among the general population, around 20% of people experience a negative response to at least some misophonic triggers [11,17], though in post-secondary populations specifically, this ranges from 23.1% and 78.9% [11,22] of participants experiencing some degree of subclinical misophonia. Clinical cases of misophonia were present in 12.2–19.9% of the post-secondary samples in the United States, China, and India [11,22,23]. In general population studies, clinical misophonia was found to affect 18% [24] 12.8% [25], 5.0% [26], and 4.6% [27] of people. Of note, many of these studies employed different measures and cutoff criteria for the distinction between clinical and subclinical sensitivities. Furthermore, many of these studies predate the recent consensus definition from Swedo and colleagues, making them difficult to compare with each other.

Hyperacusis is currently estimated to affect between 3.2% and 17.2% of the population [13,28], with a more recent study reporting hyperacusis in 26.43% of their adult sample [29]. The overlap this condition has with misophonia and other DST conditions such as tinnitus (a condition in which a person perceives sound even when there is no external source of sound, often leading to discomfort and distress) is substantial, especially in the impact and impairment domain, yet it is indeed distinct [30]. This comorbidity additionally causes difficulty in determining true prevalence figures.

## Autism

Recently, a high prevalence of DST in autistic people has been reported [2,3,5]. Sensory processing differences are a common feature of autism, and sound-based sensitivities are the most common of these [2,3,31]. As such, it is unsurprising to see a high prevalence of DST. According to recent work on this topic, 35.5% of autistic individuals have clinical misophonia [32]. However, much of the research exploring DST in autistic people suggests that misophonia and hyperacusis are not phenomenologically different or unique in autistic individuals [5], nor are the DST symptoms merely a characteristic of autism itself. Given autism is a spectrum, with non-autistic people often displaying at least some autistic traits [33], it is unclear whether non-autistic people with more autistic traits may also show a higher rate of DST. Better understanding the relationship between autistic traits and DST may provide insight into the etiology of DST.

## Gender

Sex and gender are other areas where differences in the prevalence of DST subtypes are under investigation. Prevalence has been found to be higher among women than men in some studies [17,25,27,32], and at least one study found that the severity of misophonic symptoms was higher among female survey respondents [9]. In contrast however, some studies do not find significant differences in the prevalence of misophonia between men and women [23,26,34]. Much of the research in recent years includes samples that have many more women than men, though it is uncertain whether this is due to misophonia being more common among women. Nevertheless, more research is needed to illuminate the relationship between gender and sex, and misophonia and other DST subtypes. Interestingly, while recent studies have begun

to describe participants that identify as non-cisgendered, such as transgender men and women, and nonbinary or genderfluid individuals, there appears to be no research investigating prevalence and experience of misophonia or any other DST subtypes in this population specifically.

## Current study

Taken all together, misophonia and hyperacusis may actively affect a substantial number of adults who are attending post-secondary institutions in Canada, and these conditions may have a serious negative influence on their mental and physical well-being, social, academic, and professional success, and on interpersonal relationships. As such, the goals of this study are to, (1) establish an estimate of misophonia prevalence amongst Canadian post-secondary students. Given the broad suite of self-report measures for misophonia that are currently available, and the recent adoption of the consensus definition from Swedo and colleagues [7], the current study estimated the prevalence of misophonia in Canadian undergraduate students using the Misophonia Questionnaire (MQ), the Duke Misophonia Questionnaire (DMQ), and the Duke-Vanderbilt Misophonia Screening Questionnaire (DVMSQ). Each of these iterate on previous questionnaires and have their own advantages and disadvantages. Next, (2) the Inventory of Hyperacusis Symptoms (IHS) was used to estimate the prevalence of hyperacusis in Canadian undergraduate students. By collecting a large sample of students this enabled us to, (3) investigate differences in misophonia and hyperacusis prevalence by gender and (4) explore whether more severe DST is present in students with higher autistic traits, regardless of their autistic diagnosis status. Lastly, (5) due to concern regarding the impact of social withdrawal on social skill development, we sought to investigate whether overall social competence is negatively associated with a higher severity of DST. Better understanding who is affected by misophonia and hyperacusis, and to what extent, will enable higher quality research on targeted treatments, it may generate more funding and interest within healthcare systems, and legitimize and reduce stigma for DST.

## Materials and methods

### Participants

A sample of 3096 undergraduate students (M age = 19.61, SD = 2.73, 697 male, 2275 female, 47 non-cisgendered) participated in an online survey administered through Qualtrics (Qualtrics, Provo, UT) using the research participant portal at Wilfrid Laurier University in Ontario, Canada. See Table 1 for participant demographic information for final sample. The study was advertised as investigating the quality of life of university students, with no mention of DST or related concepts, to prevent biasing the sample towards participants interested in, or experiencing, DST. Participants received course credit for their participation. All procedures were approved by the Wilfrid Laurier University Research Ethics Board and were in accordance with the World Medical Association 2013 Declaration of Helsinki.

**Table 1. Demographic information.**

| *Variable* | |
|---|---|
| N | 2080 |
| Age | 20.47 (3.35) |
| Gender | 435 male, 1600 female, 45 non-cisgendered |
| Race | White (61%) |
| | South Asian (13%) |
| | Black (6%) |
| | East Asian (4%) |
| | Other (16%) |

*Note:* Initial sample prior to data integrity check was N = 3096.

## Procedure

After consenting to participate, participants provided demographic information such as age, gender, race, etc. Next participants completed self-report measures of DST, autistic traits, and social competence. In addition, participants completed measures of anxiety, depression, and sensory processing, however, these measures are outside of the scope of the current manuscript and will thus are reported elsewhere, including in recent work by Wickie and colleagues [35]. The measures took approximately 1 hour to complete, after which respondents were debriefed and informed that the true aim of the study was to investigate DST and its relationship with autistic traits and social competence. Data collection ran from November 2023 to April 2024.

## Measures

See Table 2 for the internal consistency of all measures used in the current study where applicable and S1 Appendix for additional details on the psychometric properties of these measures, such as validity.

### Autistic traits

The Autism-Spectrum Quotient (AQ) [36] is a widely used assessment tool for measuring the presence of autistic traits. It consists of 50 items that explore a wide variety of traits common to the experience of many autistic individuals. Items such as "I prefer to do things with others rather than on my own", are assessed with a 4-point Likert scale ranging from "definitely agree" to "definitely disagree". The AQ was used to measure autistic traits present in the sample. A cutoff score of 29 or higher has been shown to reliably indicate the presence of autism [37].

### Social competence

The Multidimensional Social Competence Scale (MSCS) [38] is a 77 item self-report measure originally developed to assess overall social competence in autistic individuals, though it has been validated for use with autistic and

**Table 2. Mean scores, standard deviations, and significance on key measures and subscales by gender group.**

| | Total | | Female | | Male | | Non-Cisgendered | | Test of Significance* | Internal Consistency (Cronbach's alpha) |
|---|---|---|---|---|---|---|---|---|---|---|
| *Variable* | *n* | *M (SD)* | *n* | *M (SD)* | *n* | *M (SD)* | *n* | *M (SD)* | | |
| AQ | 2080 | 20.17 (6.57) | 1600 | 20.14 (6.60) | 435 | 19.60 (6.08) | 45 | 26.71 (6.79) | F(2, 114.8) = 22.57, p < .001, η² = 0.023. | 0.78 |
| MSCS Total | 2080 | 286.56 (31.30) | 1600 | 288.89 (31.24) | 435 | 280.53 (29.11) | 45 | 261.29 (35.20) | F(2, 114.1) = 25.41, p < .001, η² = 0.026 | 0.94 |
| MQ Total | 2080 | 18.89 (11.87) | 1600 | 20.28 (11.72) | 435 | 12.73 (9.84) | 45 | 28.84 (12.96) | F(2, 114.7) = 106.83, p < .001, η² = 0.082 | 0.90 |
| MQ Severity | 1025 | 3.16 (2.79) | 805 | 3.24 (2.82) | 200 | 2.67 (2.54) | 28 | 4.32 (3.07) | F(2, 69.3) = 6.05, p = .004, η² = 0.012 | |
| DMQ Total | 2080 | 62.63 (53.66) | 1600 | 66.74 (54.37) | 435 | 40.56 (40.08) | 45 | 125.71 (54.54) | F(2,115.8) = 95.38, p < .001, η² = 0.070 | 0.98 |
| DMQ Symptom Severity | 2080 | 22.16 (19.67) | 1600 | 23.98 (19.90) | 435 | 13.12 (14.40) | 45 | 44.47 (20.16) | F(2,115.8) = 112.82, p < .001, η² = 0.079 | |
| DMQ Coping | 2080 | 21.34 (17.85) | 1600 | 22.54 (17.98) | 435 | 15.05 (15.21) | 45 | 39.47 (15.77) | F(2, 116.6) = 69.48, p < .001, η² = 0.052 | |
| IHS Total Score | 2080 | 39.94 (15.41) | 1600 | 40.80 (15.54) | 435 | 34.85 (12.04) | 45 | 58.60 (19.12) | F(2, 114.2) = 62.17, p < .001, η² = 0.059 | 0.96 |

*Note:* Autism Spectrum Quotient (AQ); Multidimensional Social Competence Scale (MSCS); Misophonia Questionnaire (MQ); Duke Misophonia Questionnaire (DMQ); Inventory of Hyperacusis Symptoms (IHS)

non-autistic samples [39]. The MSCS measures social competence across 7 domains: social motivation, social inferencing, demonstrating empathic concern, social knowledge, verbal conversation skills, non-verbal sending skills, and emotional regulation. Items such as: "I avoid talking to people when possible (e.g., look, move, or walk away)" are assessed with a 5-point Likert scale ranging from 1 (not true or almost never true) to 5 (very true or almost always true). Items are reverse scored, with higher subscale and total scores corresponding with greater social competence.

### Decreased sound tolerance

**Misophonia Questionnaire (MQ).** The MQ [11] is a self-report measure of misophonia symptoms, behavioural responses, and severity, consisting of 3 sections. The first section, the Misophonia Symptom Scale, probes for the presence of sound triggers with the statement "In comparison to other people, I am sensitive to the sound of:" and asks whether a series of the most common trigger sounds (e.g., eating, tapping, etc.) cause a strong negative reaction in the respondent. Items are rated from 0 (not at all true) to 4 (always true). This section also includes an opportunity to add in other trigger sounds that bother the respondent but are not present on the list. In the second section, the Misophonia Emotions and Behaviours Scale, the frequency of emotional and behavioural reactions are investigated. Respondents are asked "Once you are aware of the sound(s), because of the sound(s), how often do you:" with items such as "become anxious or distressed". These are rated from 0 (never) to 4 (always). The scores are summed to form a total score, which can range from 0 to 68. In the final section, respondents are asked to rate on a 1 (minimal) to 15 (very severe) scale how severe the overall impact of their sound sensitivity is on their day-to-day life. A score of at least 7 on the severity subscale, as well as an average score of 2 or higher on the Emotions and Behaviours subscale has been suggested to indicate the presence of misophonia [40], as this cut-off indicates at least "moderate sound sensitivities" causing "significant interference".

**Duke Misophonia Questionnaire (DMQ).** The DMQ [40], is an 86-item measure of misophonia. The DMQ assesses affective, physiological, and behavioural responses to sound triggers, coping strategies, and the impact misophonia has on one's life. The DMQ is one of the most comprehensive self-report misophonia measures to date. The DMQ contains 9 subscales that measure: (1) trigger frequency (16 items), (2) affective responses (5 items), (3) physiological responses (8 items), (4) cognitive responses (10 items), (5) coping before (6 items), (6) coping during (10 items), (7) coping after (5 items), (8) impairment (12 items), and (9) beliefs (14 items). Beyond these subscales, there are also composite scales for overall *Symptom Severity* (combined affective, physiological, and cognitive subscales) and *Coping* (combined coping subscales). The types of questions vary across the subscales but mostly involve Likert scale responses. When compared to multiple other misophonia self-report measures, DMQ scores have been found to be highly correlated, indicating reliable assessment of caseness for misophonia (see [40] for details). Scores on each subscale are summed for a total score, and a cut-off of 39 or greater indicates severe symptomology, which qualifies the respondent for misophonia on this measure.

**Duke-Vanderbilt Misophonia Screening Questionnaire (DVMSQ).** The DVMSQ [32] is a 21-item measure of misophonia that complements the DMQ, assessing the symptoms and functional impairments related to misophonia. The DVMSQ also contains a diagnostic algorithm that determines whether cases are clinical or subclinical, based on the consensus definition of misophonia [7,32]. The first item is a Yes/No screening question that asks if there are specific sounds the respondent is *extremely* bothered by, even if they are not loud. If the respondent selects 'no', they are instructed to discontinue. If the respondent selects 'yes,' they continue to 20 Likert items. The Likert items include 12 "symptom frequency" items (rated on a Likert scale from 0 (never) to 4 (very often)), as well as 8 "impairment" items (rated on a scale from 0 (not at all) to 4 (an extreme amount)). The questionnaire's diagnostic algorithm, which compiles scores across all items, is then used to determine if a respondent has no misophonia, subclinical misophonia, or clinical misophonia.

**Inventory of Hyperacusis Symptoms (IHS).** The IHS [41] is a 25-item self-report measure used to assess hyperacusis symptoms. The IHS uses 25 Likert items such as "Compared to most people, common everyday sounds seem excessively loud to me" with responses ranging from "not at all" to "very much so". The questionnaire addresses symptoms, impact on daily functioning, and other relevant areas. Scores are summed from 0 to 100, with a cut-off score of 69, indicating that the symptoms cause at least "somewhat of a problem," used to suggest the presence of hyperacusis. More recent research however has suggested that a cut-off score of 56 provides a superior balance of sensitivity and specificity [42], therefore we examined prevalence in our sample by considering both cut-offs.

## Data integrity check

To ensure the quality of the data and accurate responses from participants, several measures were taken. First, a "commitment check" was inserted before the questionnaires began. It described the importance of deliberate and thoughtful responses to the survey and asked participants "Do you commit to providing thoughtful answers to all the questions in this survey?". Response options were "Yes, I will", "No, I will not.", or "I can't promise either way". Next, 4 attention check questions such as "Please select 'Often true' to confirm you are reading these questions closely." were inserted into the survey to ensure that respondents were adequately reading and attending to the content. A final measure used in this survey was an "honesty check" that informed participants that there would be no penalty for their answer. It simply asked, "Did you answer any of the above questions randomly?". Participants who answered "yes" to this question, or incorrectly responded to any of the attention check questions were removed from the dataset and analyses.

After removing low-quality data ($n = 1016$), a final sample of 2080 participants were included in the analyses for the present study (M age $= 20.47$, SD $= 3.35$; 435 male, 1600 female, 45 non-cisgendered). Additionally, MQ severity data was only collected from 1033 of these participants due to an error in the Qualtrics program.

## Data analysis

Analyses were conducted in the statistical programs Jamovi (version 2.3.28), and Jasp (version 0.19.3). Due to unequal variances, Welch's ANOVAs were used, with Games-Howell and Tukey HSD post-hoc tests used for continuous variables to determine whether scores differed significantly across gender groups, and Chi-square tests of association were used for proportions of clinical cases. Effect sizes are reported using Cohen's $d$, $\eta^2$, Cramer's $V$, and Pearson's $r$, where appropriate. To examine associations between DST variables and AQ and MSCS, Pearson's correlations were conducted. Due to a large sample size, Cohen's $d$ effect sizes greater than $+/-0.3$, $\eta^2$ effect sizes greater than 0.06, and Pearson's $r$ effect sizes greater than $+/-0.3$ were considered moderate and meaningful effects.

## Results

### Prevalence

**Misophonia.** Estimates varied between each self-report measure for misophonia (see Fig 1). For the MQ, 12% ($n = 128$) of the sample met the clinical cut-off for misophonia. This prevalence estimate was different among our gender groups, with 14% ($n = 114$) of women meeting cut-off, while 10% ($n = 20$) of men, and 25% ($n = 6$) of those with non-cisgender expression met the cut-off. Chi-square tests of association indicated the prevalence of clinical misophonia showed a weak difference across genders, $\chi^2 = 30.5$, $p < .001$, $V = 0.175$. Standardized residuals indicated that clinical cases were more prevalent for females and non-cisgendered participants, relative to male participants, but female and non-cisgendered participants were not significantly different from each other. See S2 Table for detailed statistics.

MQ Severity scores were found to be significantly different between gender groups, (F(2, 69.3) = 6.05, p = .004, $\eta^2 = 0.012$). Tukey corrected post-hoc tests using Cohen's $d$ revealed a small effect size for the difference between men and women ($d = -0.208$, $p_{Tukey} = 0.023$), while the difference between men and non-cisgendered participants demonstrated

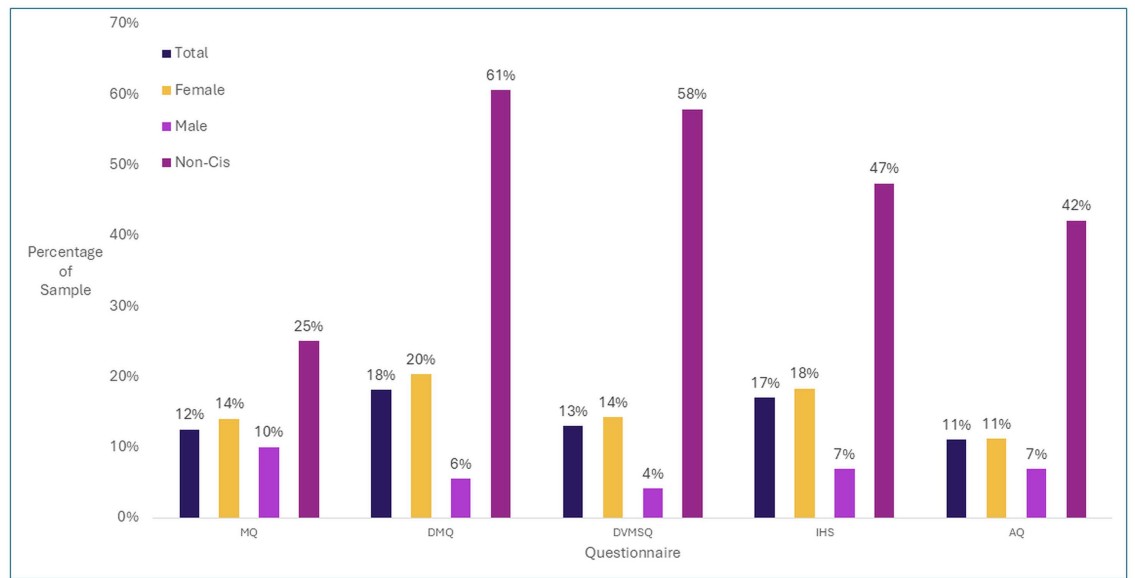

**Fig 1. Prevalence of clinical DST and autism in each gender group according to each measure.** Misophonia Questionnaire (MQ), Duke Misophonia Questionnaire (DMQ), Duke Vanderbilt Misophonia Screening Questionnaire (DVMSQ), Inventory of Hyperacusis Symptoms (IHS), Autism spectrum Quotient (AQ).

a medium effect size ($d=-0.597$, $p_{Tukey}=0.009$). The difference between women and non-cisgendered participants was not significant ($d=-0.389$, $p_{Tukey}=0.107$).

For the DMQ, misophonia was reported in 18% (n = 374) of the sample. Prevalence according to DMQ was 21% (n = 328) among women, 6% (n = 24) among men, and 61% (n = 23) among non-cisgendered participants. By gender, chi-square tests of association indicated the prevalence of clinical misophonia showed a moderate difference across genders, $\chi^2=99.7$, p < .001, $V=0.219$. Standardized residuals indicated that clinical cases were more prevalent for females and non-cisgendered participants, relative to male participants. See S3 Table for detailed statistics.

DMQ total scores also varied across the gender groups (F(2, 115.8) = 78.2, p < .001, $\eta^2=0.070$). Tukey corrected post-hoc tests using Cohen's $d$ revealed a medium effect size for the difference between men and women ($d=-0.506$, $p_{Tukey}<0.001$) while the differences between men and non-cisgendered participants, as well as women and non-cisgendered participants demonstrated large effect sizes ($d=-1.647$, $p_{Tukey}<0.001$) and ($d=-1.140$, $p_{Tukey}<0.001$) respectively for the total score. The DMQ Severity subscale scores were also significantly different across groups (F(2, 115.8) = 88.9, p < .001, $\eta^2=0.079$). Post hoc tests demonstrated a medium effect size for the difference between men and women ($d=-0.577$, $p_{Tukey}<0.001$), a large effect size for the difference between men and non-cisgendered participants ($d=-1.660$, $p_{Tukey}<0.001$), and a large effect size for the difference between women and non-cisgendered participants ($d=-1.084$, $p_{Tukey}<0.001$). Finally, DMQ Coping subscale scores were also significantly different across groups (F(2, 116.6) = 56.9, p < .001, $\eta^2=0.052$), with the men and women demonstrating a significant but small effect size difference ($d=-0.431$, $p_{Tukey}<0.001$), while the effect sizes for the differences between men and non-cis participants ($d=-1.406$, $p_{Tukey}<0.001$) and women and non-cis participants ($d=-0.974$, $p_{Tukey}<0.001$) were large.

The DVMSQ detected clinical misophonia in 13% (n = 272) of the sample. By gender, chi-square tests of association indicated the prevalence of clinical misophonia showed a weak difference across genders, $\chi^2=136.3$, p < .001, $V=0.181$. Standardized residuals indicated that clinical cases were more prevalent for females and non-cisgendered participants, relative to male participants, and the same was true for sub-clinical cases. See S4 Table for detailed statistics.

Furthermore, since this questionnaire also includes a categorization for subclinical misophonia, 14% of the sample (n = 287) were suggested to have sub-clinical misophonia. 14% of women (n = 230), 4% of men (n = 18), and 58% (n = 22) of non-cisgendered participants met criteria for clinical misophonia.

**Hyperacusis.** Using the cut-off score of 56, which has been suggested to have a superior balance of sensitivity and specificity [42], the IHS indicated that 17% (n = 344) of participants met the criteria for clinical hyperacusis. By gender, chi-square tests of association indicated the prevalence of clinical hyperacusis showed a weak difference across genders, $\chi^2 = 62.5$, p < .001, $V = 0.173$. Specifically, 18% (n = 293) of women, 7% (n = 30) of men, and 47% (n = 18) of the non-cisgendered participants met the clinical cut-off. Standardized residuals indicated that clinical cases were more prevalent for females and non-cisgendered participants, relative to male participants. See S5 Table for detailed statistics.

Considering total IHS scores, differences across gender were moderate (F(2, 114.2) = 65.5, p < .001, $\eta^2 = 0.059$). Tukey post-hoc tests using Cohen's d revealed a medium effect size for the difference between men and women ($d = -0.415$, $p_{Tukey} = 0.023$) while the difference between men and non-cis participants ($d = -1.605$, $p_{Tukey} < .001$) and women and non-cis participants ($d = -1.190$, $p_{Tukey} < .001$) demonstrated very strong effect sizes. Note that using the cut-off score of 69 proposed during scale development [41], the prevalence of hyperacusis was estimated at 6% (n = 134).

**Comorbid misophonia and hyperacusis.** In investigating the prevalence of the comorbidity of misophonia and hyperacusis, we found that 15% (n = 310) of the participants had clinically significant misophonia according to at least one of the MQ, DMQ, or DVMSQ. Participants who met criteria for either misophonia or hyperacusis made up 18% (n = 374), and those who met criteria for both conditions, indicating comorbidity, were 13% (n = 280) of the sample.

## Autistic traits

AQ scores indicate that 11% (n = 231) of the sample have clinically significant autistic traits. By gender, 11% (n = 181) women, 7% (n = 30) men, and 42% (n = 16) of the non-cisgendered participants had clinically significant autistic traits. Chi-square tests of association indicated the prevalence of clinically significant autistic traits showed a weak difference across genders, $\chi^2 = 58.5$, p < .001, $V = 0.168$. Standardized residuals indicated that clinical cases were more prevalent for females and non-cisgendered participants, relative to male participants. See S6 Table for detailed statistics.

A significant group difference was also found in total scores (F(2, 114.8) = 24.4, p < .001, $\eta^2 = 0.023$). Post-hoc tests using Cohen's d revealed a non-significant effect size for the difference between men and women ($d = -0.082$, $p_{Tukey} = 0.283$) while the difference between men and non-cis participants demonstrated a strong effect size ($d = -1.093$, $p_{Tukey} < .001$). The difference between women and non-cis participants was also strong ($d = -1.011$, $p_{Tukey} < .001$).

**DST and autistic traits.** Given the categorical nature of the misophonia classification on the DVMSQ, a one-way between-subjects ANOVA was conducted to investigate whether participants classified by no misophonia, subclinical misophonia, and clinical misophonia on the DVMSQ varied in their degree of autistic traits. AQ scores differed significantly across the DVMSQ classifications, ($F(2,2077) = 89.0$, $p < .001$, $\eta^2 = 0.079$), with a medium effect size. Tukey corrected post-hoc comparisons indicated that AQ scores were significantly higher with a large effect size for clinically significant misophonia (M = 24.5, SD = 7.1) compared to no misophonia ($d = -0.851$, $p_{Tukey} < .001$), a medium effect size for clinical misophonia relative to sub-clinical (M = 21.4, SD = 6.3, $d = 0.504$, $p_{Tukey} < .001$), and a medium effect size for no misophonia (M = 19.2, SD = 6.2) compared to sub-clinical ($d = -0.347$, $p_{Tukey} < .001$) (See Fig 2 for AQ total score by misophonia classification).

Given the continuous nature of the remaining DST questionnaires, Pearson's correlations were performed to investigate associations between autistic traits and the intensity of DST (see Fig 3). The association between the AQ scores and the MQ severity scores was found to be positive and weak ($r(1031) = 0.167$, $p < .001$) while the association between AQ scores and MQ Total scores was found to be positive and moderate ($r(2078) = 0.370$, $p < .001$). The relations between AQ scores and DMQ symptom severity ($r(2078) = 0.398$, $p < .001$) and coping ($r(2078) = 0.339$, $p < .001$) composite subscales, were weak to moderate and positive. The relation between AQ scores and IHS total scores was moderate and positive ($r(2078) = 0.465$, $p < .001$).

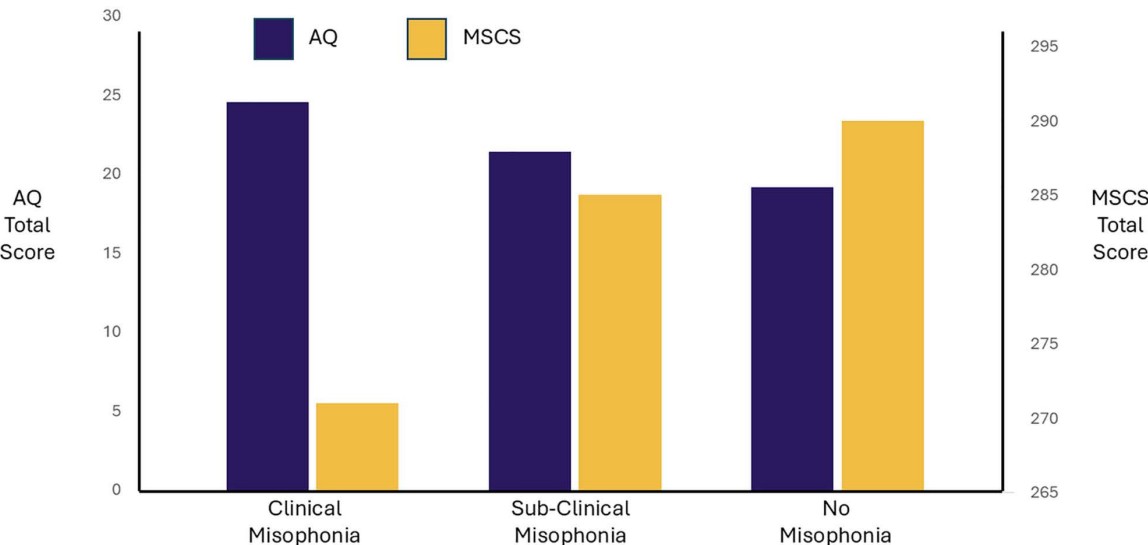

**Fig 2.  Mean Autism Spectrum Quotient (AQ) and Multidimensional Social Competence Scale (MSCS) total scores for Duke-Vanderbilt Misophonia Screening Questionnaire (DVMSQ) classifications of clinically significant misophonia, sub-clinical misophonia, and no misophonia.**

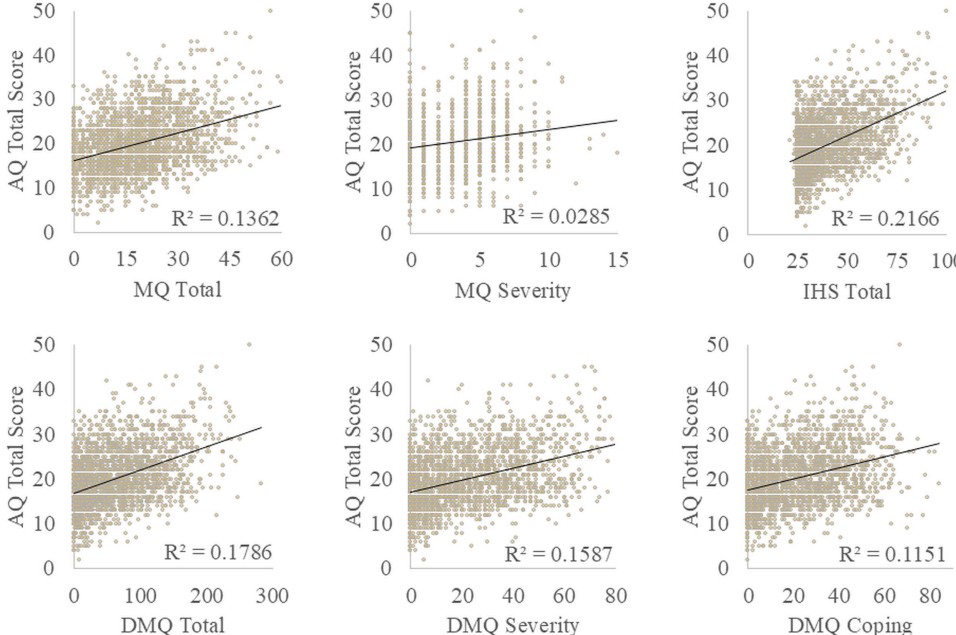

**Fig 3.  Correlation scatterplots of Autism spectrum Quotient (AQ) total scores with continuous DST measures: Misophonia Questionnaire (MQ) total, and severity subscale, Inventory of Hyperacusis Symptoms (IHS), Duke Misophonia Questionnaire (DMQ) total, severity, and coping subscales.**

## Social competence

The association between social competence, measured by MSCS scores, and DST was investigated through a series of Pearson's correlational analyses (see Fig 4). Overall social competence was found to be weakly negatively correlated with MQ severity scores ($r(1031) = -0.140$, $p < .001$) and with MQ total scores ($r(2078) = -0.294$, $p < .001$). MSCS total scores were also found to be weakly to moderately negatively correlated with the severity ($r(2078) = -0.339$, $p < .001$) and coping ($r(2078) = -0.256$, $p < .001$) subscales for the DMQ, and with the IHS total score ($r(2078) = -0.387$, $p < .001$). Finally, MSCS scores were strongly negatively correlated with AQ total scores ($r(2078) = -0.614$, $p < .001$). Associations between MSCS subscales and DST measures and AQ scores can be found in S1 Table. See Fig 2 for MSCS total scores by misophonia classification.

## Discussion

With the potential impact of DST on post-secondary success in mind, the goal of this current study was to lay a foundation for further research that may help prevent unnecessary distress associated with DST in students at post-secondary institutions. As such, the first goal of this work was to estimate the proportion of post-secondary students with clinical levels of DST through a large sample of data collected at a post-secondary institution. Following that, we also sought to better understand how DST varies across gender, and how it relates to autistic traits and social competence. Given our previous work has shown that DST promotes social avoidance and withdrawal in children [2], social competence was of interest as we theorize that DST trigger exposure leads to frequent and highly aversive negative experiences, which incentivizes avoidance of triggering environments. These avoidance and withdrawal tactics may then cause individuals to miss out on experiences that provide important academic, social, and professional development. Over an extended period, it seems logical that this may have a harmful impact on overall social development, and thus negatively affect social competence.

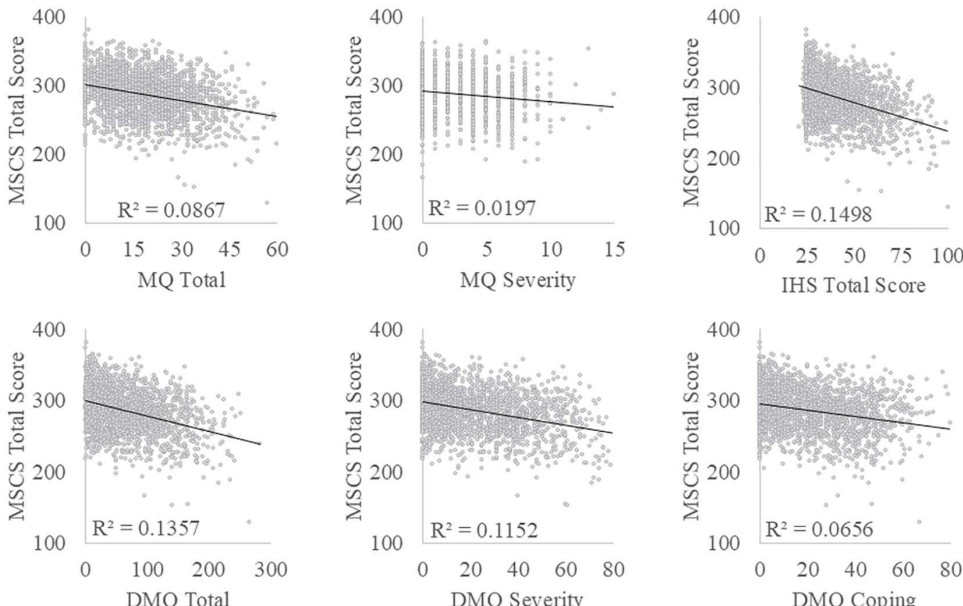

**Fig 4. Correlation scatterplots of Multidimensional Social Competence Scale (MSCS) with continuous DST measures.** Misophonia Questionnaire (MQ) Total and Severity, Inventory of Hyperacusis Symptoms (IHS) total, Duke Misophonia Questionnaire (DMQ) Total, Severity, and Coping subscales.

In the present study, the use of the DVMSQ was highly advantageous as it allowed for a prevalence estimate using a diagnostic algorithm for misophonia that takes into account the criteria outlined by the recently published consensus definition [7]. With a measure that reflects this definition, our prevalence estimates thus reflect the most up-to-date definition of misophonia. The data from this measure suggests that 13% of the post-secondary students in our sample qualify for clinical misophonia, while an additional 14% of students fall in the range of sub-clinical misophonia. Given the only distinction between the clinical and subclinical classification is the level of impairment, this suggests that approximately 27% of post-secondary students are experiencing distress as a result of misophonia triggers. Additionally, making use of the DMQ and the MQ, clinical misophonia in post-secondary students is estimated at 18%, and 12%, respectively. While overall these findings suggest that misophonia affects between 12 and 27% of post-secondary students, this variability underscores the importance of establishing robust tools to determine diagnosis and clinical cut-off criteria for misophonia. Similarly, hyperacusis prevalence varies substantially without objective diagnostic tools. Using the criteria originally established for the IHS, 6% of the sample is suggested to have clinical hyperacusis, however when using the more recently validated criteria for this measure, shifting the cutoff from a score of 69 to a score of 56, the prevalence was instead much higher at 17%. Furthermore, when considering participants who qualify for clinical levels of either misophonia on any of our measures, or hyperacusis, we found this number to be 18% of the sample, and 13% of the sample qualified as comorbid for both misophonia and hyperacusis. Regardless of the specific measures and cut-offs used, these results suggest that a significant number of Canadian university students are actively experiencing misophonia and/or hyperacusis symptoms. Given misophonia has been linked to anxiety [5; 16], depression [4], and social isolation [2], it is clear that DST is a significant and urgent health concern at post-secondary institutions.

With conflicting evidence pertaining to the relationship between gender and DST [25–27,32,34], the current study sought to examine potential gender differences in the prevalence of DST. Across all DST measures, a significantly higher percentage of women in the sample qualified for clinical DST conditions. This is in line with past studies that suggested an increased prevalence of DST in women [17,25,27,32]. The prevalence of misophonia and hyperacusis specifically among non-cisgendered participants was also explored. The data revealed a high prevalence of DST conditions among trans men and women, nonbinary, and genderfluid participants. Notably however, our total sample only contained 45 non-cisgendered students. Given that this group was comprised of highly varied gender identities that do not necessarily belong in a single statistical group like this, it would be inappropriate to draw any conclusions or state that DST is substantially more common among non-cisgendered people at this time. Nonetheless, these findings warrant further research specifically investigating DST among these different genders. Effect sizes observed in the investigation of the statistically significant relationships between gender groups were varied depending on the measure being discussed. Between men and women, most of these were small or medium. Between men and non-cisgendered participants, and women and non-cisgendered participants, these were often medium or large, however it is still important to note that the non-cisgendered group was quite small. This suggests that the relationships in this analysis have meaningful practical significance and are worth further attention.

Recent work has highlighted the association between DST and autism [2,3,5,32]. Given the broader autism phenotype extends into the non-autistic population [33], we explored the relation between autistic traits and DST. Based on our classification of misophonia using the DVMSQ, our data suggest that highest AQ scores, and thus the most autistic traits, are present in individuals with clinical levels of misophonia. Further, those with sub-clinical misophonia also reported more autistic traits than those without misophonia. Consistent with this, our measures that provided an estimate of overall misophonia (MQ, DMQ) and hyperacusis (IHS) symptom severity revealed that higher autistic traits were related to higher misophonia and hyperacusis symptom severity. This finding is in keeping with previous research that also found overall severity for both misophonia and hyperacusis to be correlated with higher autistic traits [5]. The effect sizes in these relationships ranged from medium to large, suggesting these findings are practically significant and meaningfully worth evaluating. Furthermore, DST has been found to be more highly prevalent in autistic people than in non-autistic people

[32]. This information highlights autistic post-secondary students as a group for which DST-based accommodations are especially important. As previously discussed, many universities and post-secondary institutions already make efforts to accommodate and support autistic students with sensory differences, though a formal diagnosis is often required to access such services. Given that DST appears to be more prevalent in students with subclinical autistic traits, this speaks to the need for more liberal accommodation of sensory differences, such as DST, for post-secondary students. Thus, leveraging these prevalence estimates, our hope is post-secondary institutions will enhance and improve the kinds of supports they offer to support sensory differences among both autistic and non-autistic students.

As students transition into post-secondary environments, this change often comes with a host of new social experiences and demands. Past research has shown a link between social competence and academic progress in children and adolescents [43,44], though there is a paucity of research specifically investigating whether poor social skills negatively impact academic success at the post-secondary level. We speculate however that one's social competence is likely related to overall success in post-secondary environments both academically and socially. Our results indicated that both higher misophonia and hyperacusis symptoms were associated with lower social competence, with generally small but relevant effect sizes. Given that people with misophonia and hyperacusis frequently adopt avoidance strategies to cope with their decreased tolerance to sound triggers [2], and many of the environments that are fundamental for academic and social success are likely to be filled with potentially triggering stimuli, it is possible that the measures taken to cope with DST are at the same time negatively impacting social competence. However, given these findings are correlational and observational in nature, this relationship warrants further investigation. Nevertheless, these findings suggest that many of the students who have the more severe expressions of DST appear to also have overall lower social competence, and the kinds of experiences that might improve those social skills are also difficult to safely engage in due to the risk of experiencing the negative effects of their DST condition.

Many Canadian universities and post-secondary institutions have departments or offices that provide supports to improve quality of life and chances at academic success for students with physical and mental health conditions, disabilities, sensory or developmental differences, and any other life factors that may infringe on a student's ability to succeed. It is important that disorders of DST, such as misophonia and hyperacusis, are known to and legitimized by these entities, so that students with these conditions can have supports made available to them. This may include options such as preferred seating in lecture halls, allowances to wear ear plugs or noise cancelling headphones during classes, access to note-taking services, etc. Further research should be undertaken to investigate the best accommodation techniques and the most efficient ways to implement them, especially given that resources are often spread thin in these departments, with issues like long wait times for support being an issue for some universities [21]. To that end, our team intends to explore in future research the specific ways that learning and memory are impeded as consequences of exposure to DST triggers; something highly relevant to student success at post-secondary institutions. Specific data on this would likely be of interest to university organizations committed to maximizing success for all students.

The present research represents a significant step forward in DST research in a few notable ways. First, this is among the largest samples reported thus far in the DST literature. It also represents one of the first DST prevalence estimates in a Canadian university population, and one of the first studies to make use of the DVMSQ, thus incorporating the new consensus definition. This adds to the global effort to study the populations most affected by these conditions. Given the large sample, it was possible to examine relations with gender as well as autistic traits and social abilities. However, this study is not without limitations. The most significant limitations are the use of self-report measures and the cross-sectional nature of the study. While this may impose bias in the results, there are currently no objective tools available to measure or assess DST conditions. This study was also conducted as an online survey; thus, we were unable to monitor participants as they completed the study. This limitation was mitigated by the aggressive use of attention checks and honesty measures in the study to ensure the data used in our analyses was of good quality. An additional limitation of the current study is the disproportionate number of women respondents. While this resulted in an unequal gender distribution, given

our large sample, it was still possible to explore proportional differences in DST across different genders. It is also worth noting once again that the non-cisgendered subgroup is quite small in comparison to the size of the entire sample. A final limitation is the reduced size of the sample that completed the MQ severity measure, compared to the sample that completed the other measures used in this study. Despite this missing data, a large sample size was nevertheless obtained for that measure.

## Conclusion

In summary, these findings demonstrate that clinical levels of misophonia and hyperacusis affect many Canadian university students, particularly female and non-cisgendered students. Given the prevalence findings, it is likely that nearly any lecture, tutorial, or lab will contain at least some students who actively are affected by DST symptoms. Thus, it would be beneficial to increase general awareness of conditions such as misophonia and hyperacusis in order to enact changes that may ease the burden on those affected. Simple changes that enable students with DST conditions to be exposed less often to triggering stimuli, such as limiting use of food and beverages in certain lecture halls, and/or providing or allowing active noise-canceling audio devices can potentially improve the learning experience for these students.

These conditions may lead to difficulty succeeding in social, academic, and professional contexts, all three of which are important in the lives of young adults attending post-secondary institutions, thus, something as simple as awareness of these trends may enable supports and accommodations to be more effectively applied. Our findings additionally highlight the relations between DST and autistic traits and social competence, providing additional clarity and context for the factors that influence the lives of those affected, and suggesting that more research will provide even more potential benefits. Further, the potential implications of the way women and non-cisgendered individuals appear more strongly affected by these conditions necessitates special attention. For these reasons, it is crucial that more advancements be made in our knowledge of who is affected the most by these conditions and why, so that techniques can be developed and implemented to ease the burden.

## Supporting information

**S1 Table. Pearson's correlations between MSCS subscales and each other measure used.**
(PDF)

**S2 Table. Chi-square test of association for gender and Misophonia Questionnaire diagnosis.** Note * indicating Z scores ±1.96 that demonstrate statistically significant differences. Note: Reduced sample size due to survey error.
(PDF)

**S3 Table. Chi-square test of association for gender and Duke Misophonia Questionnaire diagnosis.** Note * indicating Z scores ±1.96 that demonstrate statistically significant differences.
(PDF)

**S4 Table. Chi-square test of association for gender and Duke-Vanderbilt Misophonia Screening Questionnaire diagnosis.** Note * indicating Z scores ±1.96 that demonstrate statistically significant differences.
(PDF)

**S5 Table. Chi-square test of association for gender and Inventory of Hyperacusis Symptoms diagnosis.** Note * indicating Z scores ±1.96 that demonstrate statistically significant differences.
(PDF)

**S6 Table. Chi-square test of association for gender and Autism Quotient diagnosis.** Note * indicating Z scores ±1.96 that demonstrate statistically significant differences.
(PDF)

**S1 Appendix. Psychometric Properties of Study Measures.**
(PDF)

**S1 Data. Data for Submission.**
(PDF)

## Author contributions

**Conceptualization:** Nichole E. Scheerer.

**Data curation:** Silas Manning, Natalia Van Esch, Nichole E. Scheerer.

**Formal analysis:** Silas Manning, Nichole E. Scheerer.

**Funding acquisition:** Nichole E. Scheerer.

**Investigation:** Nichole E. Scheerer.

**Methodology:** Nichole E. Scheerer.

**Project administration:** Nichole E. Scheerer.

**Resources:** Nichole E. Scheerer.

**Software:** Nichole E. Scheerer.

**Supervision:** Nichole E. Scheerer.

**Visualization:** Silas Manning, Natalia Van Esch, Nichole E. Scheerer.

**Writing – original draft:** Silas Manning.

**Writing – review & editing:** Natalia Van Esch, Nichole E. Scheerer.

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
