## [Decision Letter · Decision Letter 0]

9 Sep 2025

Dear Dr. Scheerer,

Thank you for submitting your manuscript to PLOS ONE. After careful consideration, we feel that it has merit but does not fully meet PLOS ONE’s publication criteria as it currently stands. Therefore, we invite you to submit a revised version of the manuscript that addresses the points raised during the review process.

We look forward to receiving your revised manuscript.

Kind regards,

Zypher Jude G. Regencia, Ph.D.

Academic Editor

PLOS ONE

Journal Requirements:

2. Please update your submission to use the PLOS LaTeX template. The template and more information on our requirements for LaTeX submissions can be found at http://journals.plos.org/plosone/s/latex .

[Wilfrid Laurier University     http://dx.doi.org/10.13039/100012531     Research Support Fund    Dr Nichole Scheerer].

5. In the online submission form you indicate that your data is not available for proprietary reasons and have provided a contact point for accessing this data. Please note that your current contact point is a co-author on this manuscript. According to our Data Policy, the contact point must not be an author on the manuscript and must be an institutional contact, ideally not an individual. Please revise your data statement to a non-author institutional point of contact, such as a data access or ethics committee, and send this to us via return email. Please also include contact information for the third party organization, and please include the full citation of where the data can be found.

6. We note that you have referenced ([35] Wickie A, Van Esch N, Scheerer NE, Social success in a noisy world: Exploring the relationship between decreased sound tolerance and social profiles. Unpublished.) which has currently not yet been accepted for publication. Please remove this from your References and amend this to state in the body of your manuscript: (ie “Bewick et al. [Unpublished]”) as detailed online in our guide for authors

7. Please upload a new copy of Figures 1, 2, 3, and 4 as the detail is not clear. Please follow the link for more information: https://blogs.plos.org/plos/2019/06/looking-good-tips-for-creating-your-plos-figures-graphics/

8. Please include a separate caption for each table in your manuscript.

Reviewers' comments:

Reviewer's Responses to Questions

**Comments to the Author**

1. Is the manuscript technically sound, and do the data support the conclusions?

Reviewer #1: Yes

Reviewer #2: Yes

Reviewer #3: Yes

2. Has the statistical analysis been performed appropriately and rigorously?

Reviewer #1: Yes

Reviewer #2: Yes

Reviewer #3: Yes

3. Have the authors made all data underlying the findings in their manuscript fully available?

Reviewer #1: Yes

Reviewer #2: Yes

Reviewer #3: Yes

4. Is the manuscript presented in an intelligible fashion and written in standard English?

Reviewer #1: Yes

Reviewer #2: Yes

Reviewer #3: Yes

Reviewer #1: Reviewer comment and suggestions

Generally the findings indicate that misophonia, hyperacusis, and autistic traits are more prevalent among females and non-cisgendered individuals, with significant associations among these variables. While statistically significant, the effect sizes for group differences are often small to moderate, suggesting that individual variability plays a substantial role. Future research could benefit from longitudinal designs, clinical diagnoses, and controlling for additional confounders to deepen understanding.

• The title author need to be revised and study design should be appeared on top of the tittle

• The results you have presented demonstrate thorough statistical analyses and reveal meaningful associations among misophonia, hyperacusis, autistic traits, and social competence across gender groups. They contribute valuable insights to the existing literature. However

• Clarify Limitations: Clearly acknowledge the reliance on self-report measures, the cross-sectional nature, and potential sample biases, especially regarding the small non-cisgender subgroup.

• Strengthen Discussions: Emphasize the practical significance of effect sizes, not just statistical significance, and discuss implications cautiously given the observational design.

• Ensure Clarity & Detail: Make sure all statistical methods are transparently reported, and figures/tables are well-organized for clarity.

• Supplementary Materials: Provide detailed supplementary analyses, such as residuals and additional subgroup analyses, to enhance transparency.

• If these considerations are addressed, the study appears suitable for publication.

Reviewer #2: The conclusion is too thin. The author(s) must include other information here not just the summary of the findings. For example, the implication of the study in terms of practice and theory. Also the way it expands our understanding of the concepts and contexts of those that suffer the conditions.

Reviewer #3: Dear authors, this is a well-written article. My comments are very minor and technical:

1. Line 247: including... ? seems like the sentence did not finish.

2. Methodology: Suggest to report the reliability and validity values of each instrument

**Do you want your identity to be public for this peer review?** For information about this choice, including consent withdrawal, please see our Privacy Policy

Reviewer #1: No

Reviewer #2: No

Reviewer #3: No

---

## [Author Response · Author response to Decision Letter 1]

25 Sep 2025

Please see the attached document with responses for the reviwers.

---

## [Editor Report · Decision Letter 1]

2 Oct 2025

Decreased sound tolerance in a Canadian University Context: Associations with autistic traits, social competence, and gender in an undergraduate sample

PONE-D-25-05421R1

Dear Dr. Scheerer,

We’re pleased to inform you that your manuscript has been judged scientifically suitable for publication and will be formally accepted for publication once it meets all outstanding technical requirements.

Kind regards,

Zypher Jude G. Regencia, Ph.D.

Academic Editor

PLOS ONE
---

## [Editor Report · Acceptance letter]

PONE-D-25-05421R1

PLOS ONE

Dear Dr. Scheerer,

I'm pleased to inform you that your manuscript has been deemed suitable for publication in PLOS ONE. Congratulations! Your manuscript is now being handed over to our production team.

Kind regards,

on behalf of

Dr. Zypher Jude G. Regencia

Academic Editor

PLOS ONE